# Lead in traditional eyeliners: An investigation into use and sources of exposure in King County, Washington

Aesha Mokashi[1,2,3], Katie M. Fellows[1,2,3], Roger J. Chin[2,3], Stephen G. Whittaker[2,3], Christopher D. Simpson[1], Diana M. Ceballos[1*]

1 Department of Environmental and Occupational Health Sciences, University of Washington School of Public Health, Seattle, Washington, United States of America, 2 Hazardous Waste Management Program in King County, Seattle, Washington, United States of America, 3 Public Health – Seattle & King County, Seattle, Washington, United States of America

* dmco25@uw.edu

## Abstract

Traditional eyeliners contain high concentrations of lead, a neurotoxic metal. In King County, Washington (WA), United States (US), several refugee and immigrant populations continue to use these products, some containing up to 800,000 parts per million (ppm) lead. Lead is toxic, especially in young children where it can cause neurological defects and growth delays. We investigated lead in eyeliners used by these communities, including traditional and nontraditional products. We conducted community-based participatory research (CBPR) to understand the use of traditional eyeliners in the local Afghan community. Our study revealed that Afghan community members were largely unaware of the dangers of lead in traditional eyeliners, but once made aware, most were willing to change their behaviors. We found that traditional eyeliners contained significantly higher median lead concentrations (10 ppm) than nontraditional eyeliners (0.06 ppm). Eyeliners manufactured in the US and the European Union (EU) had significantly lower median lead concentrations (0.94 ppm) than those from Afghanistan (29 ppm) and other low- and middle-income countries (LMICs) (2.8 ppm). Some traditional eyeliners labeled as "lead-free" contained very high lead concentrations (up to 610,000 ppm), well above the US Food and Drug Administration's (FDA) guideline of 10 ppm and the WA safety guideline of 1 ppm. Our findings suggest that traditional eyeliners currently used in the US still contain hazardous materials.

## 1 Introduction

### 1.1 Uncovering the issue

In-home investigations into the sources of lead exposure in local Afghan refugee children by the Hazardous Waste Management Program (Haz Waste Program) in King

**Data availability statement:** Data for all products tested is available in Supporting information. Data is also publicly available at the "Lead Content of Consumer Products tested in King County, Washington" open data platform https://data.kingcounty.gov/d/i6sy-ckp7.

**Funding:** This work was supported by Public Health- Seattle and King County and the Hazardous Waste Management Program fund, administered by the Hazardous Waste Management Program in King County, Washington in the form of a salary for AM, KMF, RJC, and SGW. Funding for the community product lead testing events was through the King County Best Starts for Kids levy and the Hazardous Waste Management Program Fund in the form of a salary for AM, KMF, RJC, and SGW. The specific roles of these authors are articulated in the 'author contributions' section. The funders had no role in study design, data collection and analysis, decision to publish, or preparation of the manuscript.

**Competing interests:** The authors have declared that no competing interests exist.

County, Washington (WA), United States (US) in 2021 revealed that some traditional eyeliners from Afghanistan contained high lead concentrations [1]. The in-home investigations were prompted by data provided by the WA Department of Health (WA DOH), which reported that Afghan refugee children aged 0–16 years had the highest prevalence of blood lead levels (BLLs) greater than the Centers for Disease Control and Prevention's (CDC) blood lead reference value (BLRV) compared to all other refugee children in WA (2016–2020 data) (Personal communication between Azadeh Tasslimi (WA DOH) and Stephen G. Whittaker (Haz Waste Program), 2021). WA DOH's data revealed that 44% of newly resettled Afghan children in WA had BLLs greater than 5 micro grams of lead per deciliter of blood (µg/dL) (CDC's BLRV in early 2021).

Additional local product data was gathered in 2022–2023, when Public Health-Seattle & King County (PHSKC) and the Haz Waste Program convened community product lead testing events in collaboration with several community-based organizations (CBOs) [2,3]. Community members were invited to bring products that could potentially contain lead to these events. The widespread use of lead-containing traditional eyeliners by our local newcomer communities was confirmed using on-site X-ray fluorescence (XRF) analyzer product screenings and subsequent laboratory analyses [3]. These events yielded eight traditional eyeliners that contained lead concentrations between 390,000 parts per million (ppm) and 840,000 ppm [3]. A public online report summarizes the 2022 product testing data [3]. All PHSKC product testing data is available on the "Lead Content of Consumer Products tested in King County, Washington" open data platform [2].

## 1.2 Traditional eyeliners – history and cultural significance

Traditional eyeliners have been used for thousands of years by cultures worldwide to adorn, treat, and protect the eye. These products are referred to by different names, including "kohl," "kajal," and "surma," and have been demonstrated to contain toxic ingredients, including lead [2,4–13]. However, communities throughout Africa, South Asia, the Middle East, and elsewhere continue to use these eyeliners, where they are frequently applied to children. The use of traditional eyeliners dates back to the ancient Egyptians [14]. Additionally, holy Islamic texts describe the Prophet Muhammad wearing eyeliner daily, and followers are encouraged to do so for religious practices and to improve eye health [15,16]. Consequently, the use of these products may be deeply rooted in tradition, religion, and culture. For example, Afghan parents often apply surma to their children to emphasize the beauty of the eyes, ward off "the evil eye," or provide medicinal benefits [13]. Studies in Pakistan found that 14.3% of new mothers applied traditional eyeliner to their children because they had been counseled by older women in their families to do so [17,18].

Different terms used for traditional eyeliners may refer to different products containing a wide variety of ingredients, ranging from soot (i.e., from burning plant material or animal fats) to powdered lead sulfide (i.e., galena). Confusion about terminology is widespread in the scientific literature, at least partially because the same

types of products are assigned different names in different countries and cultures—in some cases, multiple names may refer to the same product.

### 1.3 Traditional eyeliners as a source of lead exposure

Traditional eyeliners are applied to the mucous membrane of the eye and on the exterior of the eyelid. Adults and children use these products, but children are particularly vulnerable to lead exposure due to their propensity for hand-to-mouth behavior [19]. While ingestion is a potentially significant route of exposure, there is little information on the potential for lead to enter the bloodstream via the mucous membranes of the eye [20].

There is no safe level of lead exposure [21]. In children, lead's impact on neurological development is of particular concern [19]. Adverse neurological health effects in children exposed to lead include changes in mood, decreased learning ability, deficiencies in neuromotor and neurosensory functions, and encephalopathy [19]. Lead may also interfere with development in the womb, leading to decreased birth weight and size in infants [19]. In adults, low-level chronic exposure to lead can also result in ischemic heart disease, heart rate variability, stroke, and peripheral artery disease [22].

Although epidemiological studies have found a significant association between the application of traditional eyeliners and high lead exposures in young children, factors such as detection bias, misclassification, and other confounding variables (e.g., lead in soil and drinking water) may impact the findings [20,23–25]. However, several clinical cases in the US have associated childhood lead exposures with the use of traditional eyeliners [26–28]. In 2011, lead poisoning in a 6-month-old infant in Boston was associated with a traditional eyeliner purchased in Nigeria that contained 826,000 ppm lead [26]. In 2013, the New Mexico Department of Health identified two children who had immigrated from Afghanistan with BLLs of 27.0 µg/dL and 33.5 µg/dL [27]. The lead exposures were attributed to the use of surma originating from Afghanistan. In 2024, the New York City Department of Health and Mental Hygiene reported that lead poisoning in a mother and her four children in New York City was associated with the use of surma, which contained 390,000 ppm lead [28].

### 1.4 Present study

Our initial finding of lead-containing traditional eyeliners in the homes of local lead-exposed Afghan children, coupled with finding additional potentially hazardous cosmetics from our community product lead testing events, prompted us to conduct a more extensive investigation into the cultural basis for their use and the potential for lead exposures.

Therefore, the goals of this current study were to 1) learn if the use of traditional eyeliners reflects cultural and religious practices, 2) further characterize the products used by our local communities and the terminology used to describe them, 3) conduct lead analysis on samples of both traditional eyeliners and alternatives that our local communities consider to be safe, and 4) determine the characteristics of eyeliner products associated with high lead concentrations.

## 2 Materials and methods

### 2.1 Learning from the community

A community-based participatory research (CBPR) approach was employed to learn more about the cultural and religious practices underlying the use of eyeliners in the local Afghan community in King County, WA, the types of eyeliners they used, and opportunities to promote the use of safer alternatives. Incorporating CBPR principles ensured active involvement and engagement from community members as co-creators to foster a culturally appropriate study that addressed community-led concerns [29,30].

This CBPR approach was essential because we value the knowledge and lived experiences of community members. The goal was to ensure that the outcomes were relevant and improved the health of the Afghan community. In this study, community health advocates from the Afghan Health Initiative (AHI), a local CBO, were actively involved in data collection.

The community health advocates were fluent in Afghan languages and understood community cultural practices, which promoted greater integrity and fostered reassurance among community members regarding the research.

The first phase entailed co-creating a comprehensive survey with AHI, which was then administered by a community health advocate in the homes of 32 local families. Recruitment and administration occurred between June 1 and September 30, 2023. The in-home survey comprised 30 questions about the cultural and religious reasons community members used traditional eyeliners, the current eyeliners they were using, barriers to switching to safer alternatives, and suggestions for promoting safer products. The process included multiple-choice and open-ended questions to understand the community members' experiences and offer insights into their concerns. Data was collected on an iPad using electronic survey software to improve accuracy and efficiency while minimizing data entry errors. The community health advocate also gathered eyeliner samples used by the survey participants for chemical analysis. Community members were provided with an informed consent form that clearly outlined the voluntary nature of their participation, their right to withdraw at any time, and the assurance that opting out would not affect the services provided to them. To facilitate understanding, community health advocates were trained to interpret the consent form if necessary and address any questions from community members. Verbal consent was obtained from community members, which was documented in the data collection software by the community health advocate.

The second phase involved a community forum in South King County, WA - an area with a large Afghan population - on November 18, 2023. Recruitment for this event occurred from September 30 to November 11, 2023. Twenty-nine Afghan community members engaged in a discussion to gain additional insights into lead in traditional eyeliners. The community forum served to verify and complement information gathered from the in-home surveys to ensure data accuracy and reliability.

The Haz Waste Program conducted training sessions for AHI community health advocates, focusing on data collection and group facilitation techniques. The community health advocates were assigned either a facilitator or a note taker role. They were then assigned to a community group based on their proficiency in Afghan languages and understanding of cultural norms. Notetakers were instructed to document group dynamics, engagement levels, nonverbal cues, and noteworthy moments that could reveal sentiments often overlooked in audio recordings.

Community members were organized into culturally appropriate groups according to their preferred identities and language to ensure they felt comfortable interacting and sharing information with other group members. Given the extensive feedback from community members, permission to record discussions was received via an informed consent form. Community health advocates were trained to interpret the consent form if necessary and address any questions from community members. Community members were informed that their verbal consent was required, and a tape recorder was visibly placed in the open, ensuring they were aware that the session was being recorded. This transparency ensured that their participation was voluntary. Consent was documented in the data collection software by the community health advocate. The community health advocates also gathered eyeliner samples used by the forum participants for chemical analysis. AHI staff members translated the recordings into English for analysis.

For each phase, demographic information was collected, including geographic location, preferred language, age range, gender identity, education level, household structure, employment status, years spent in the US, and estimated annual household income. In addition, Haz Waste Program staff were engaged in ongoing conversations with AHI staff, who provided considerable insights into eyeliner use by their community. The Haz Waste Program provided funding to AHI and financial compensation to participating community members. Funding was provided by the Hazardous Waste Management Program fund.

## 2.2 Ethics statement

This public health activity is not subject to review by the Institutional Review Board (IRB) because the scope is limited to public health practice, and all activities are authorized and conducted by PHSKC, a public health authority. Although this

study did not require IRB approval, we took appropriate measures to uphold ethical standards by obtaining formal verbal consent from all participants and ensuring the safety and protection of our community members. We did not work with minors, and parental or guardian consent was not applicable.

## 2.3 Acquisition of eyeliner samples

A total of 145 eyeliners, including traditional and nontraditional products, were collected for analysis between 2022 and 2023 from four different sources: 1) the community product lead testing events, 2) collaborations with the local Afghan community, 3) the in-home investigations of lead-exposed Afghan children, and 4) products purchased online by the Haz Waste Program (from Amazon, Etsy, and eBay) that were recommended by the local community. Products purchased online included the top results that came up when searching the terms "kajal", "kohl", and "surma", in addition to alternative products considered acceptable by the Afghan community. "Kajal", "kohl", and "surma" were used as search terms because these eyeliners were consistently identified as having high lead concentrations during in-home lead investigations and community product lead testing events. We aimed to collect or purchase at least one gram (1 g) of sample to prevent matrix interference during laboratory analysis and achieve reporting limits (RL) below 1 ppm. The number of samples collected by each method is listed in Table 1. More details on sample acquisition are provided in S1 Appendix.

## 2.4 Characterization of eyeliner samples

Information for each sample was gathered from labels on the eyeliner, the website the eyeliner was purchased from, and/or the community member who donated the product. The following information was recorded: traditional or nontraditional product, assigned product type (i.e., kajal, kohl, or surma), whether it was homemade or manufactured, country of production, brand and manufacturer, texture (i.e., powder vs. cream), disclosed ingredients, and whether the product was labeled as "lead-free."

## 2.5 Sample analysis

Once eyeliners were acquired, a 210-millimeter (mm) plastic spatula was used to transfer 1 g of eyeliner product from its original container into a plastic sample bag. Samples were then submitted for chemical analysis to two laboratories: the University of Washington's Environmental Health Laboratory and Trace Organics Analysis Center (UW EHL TOAC, Seattle, WA) and NVL Laboratories, Inc. (Seattle, WA). Samples were prepared and analyzed following published methods. Analysis conducted by Inductively Coupled Plasma-Mass Spectrometry (ICP-MS) was based on the US Environmental Protection Agency (EPA) Method 6020a [31]. Analysis conducted by Inductively Coupled Plasma Optical Emission Spectroscopy (ICP-OES) was based on the US Consumer Product Safety Commission Directorate (CPSC)-CH-E1002-08 (Children's non-metal products) Method [32]. Analysis conducted by Graphite Furnace Atomic Absorption Spectrometry (GFAA) was based on US EPA Method 7010 [33]. RL based on 1 g of sample were between 0.05 ppm and 4 ppm, depending on the analytical method used. The employment of different analytical methods arose from the use of pre-existing data

**Table 1. Source of eyeliner samples.**

| Source | Traditional eyeliners | Nontraditional eyeliners | Total |
|---|---|---|---|
| Community product lead testing events | 24 | 39 | 63 |
| AHI collaborations | 21 | 0 | 21 |
| In-home lead investigations | 4 | 0 | 4 |
| Purchased online | 31 | 26 | 57 |
| Total | 80 | 65 | 145 |

AHI, Afghan Health Initiative.

collected by PHSKC during in-home lead investigations and community product lead testing events. However, once our study commenced, all eyeliner samples were analyzed exclusively using GFAA because NVL Laboratories, the primary laboratory conducting eyeliner analysis, was able to achieve the lowest RL with GFAA. In contrast, eyeliners tested before this study were analyzed using multiple methods, as different PHSKC programs submitted samples for testing without a standardized analytical protocol in place at the time. Analytical details and quality control parameters are provided in S1 Appendix.

## 2.6  Statistical analysis

Qualitative thematic analysis for the in-home surveys and the community forum survey was conducted using ATLAS.ti Version 23 and Microsoft Excel Version 2402. Reflexive thematic analysis was used to maintain the anonymity of Afghan community members [34]. Community members provided nuanced and multifaceted responses, and reflexive thematic analysis has been proven effective in capturing the key themes and broader concepts being expressed [34]. We conducted an iterative and structured process of coding and categorizing responses to discern prevalent trends and patterns.

Quantitative statistical analyses were used to determine eyeliner characteristics associated with higher lead concentrations. Descriptive statistical analyses were conducted using Microsoft Excel Version 2405 and RStudio Version 2024.04.0 (Build 735). Bivariate and multivariate analyses were performed using nonparametric tests to determine whether there were significant differences ($p < 0.05$) in lead concentrations based on the characteristics described in Table 2. Because the data were not normally distributed, we employed the two-sided Wilcoxon Rank-Sum test for bivariate analysis. Similarly, the Kruskal-Wallis test was employed for multivariate analysis. To ensure the analyses had sufficient statistical power, we confirmed that each variable was represented by six or more samples. For samples with lead concentrations <RL, laboratory-reported lead concentrations were included in the analysis when available. However, samples with lead concentrations reported as 0 ppm were included in the analysis using the NADA2 and EnvStats packages in R. Specifically, left-censored imputations were conducted with maximum likelihood values estimated for censored observations [35]. Regression analysis results and reporting parameters can be found in S2 Appendix. Eyeliner characteristic data can be found in S3 Appendix.

## 3  Results

### 3.1  Learning from the community

Most participants indicated they used traditional eyeliners, including surma, kajal, and kohl. Approximately 33% reported frequent use of these products (i.e., 16–30 times over 30 days), 22% reported regular use (i.e., 6–15 times over 30 days), and 33% reported occasional use (i.e., 1–5 times over 30 days). Approximately 3.2% reported never using these products, and the rest (7.5%) were unsure. The use of traditional eyeliners extends beyond women and children, with men also using them.

Many of the responses indicated that the use of traditional eyeliners is deeply rooted in culture and tradition. We found that a significant factor behind their use appears to be the influence and recommendations of family members and respect for cultural customs. Some community members believe traditional eyeliners are beneficial for eye health, making the eyes stronger and more attractive. Others believe traditional eyeliners are more "natural" than nontraditional, manufactured eyeliners. One participant told us they use traditional eyeliners for religious reasons.

Approximately 59% of the participants confirmed their children used traditional eyeliners. Out of those children, 42% used traditional eyeliners frequently (16–30 times over 30 days), 21% used them regularly (6–15 times over 30 days), and around 26% used them occasionally (1–5 times over 30 days).

We learned that Afghans refer to two types of surma. While "oily surma" is made by combining soot and animal fat, "stone surma" is made by grinding minerals, including galena (Personal communication between Reza Pedram (AHI),

**Table 2. Eyeliner characteristics considered in the statistical analysis.**

| Eyeliner characteristic[a] | Variable | Rationale |
|---|---|---|
| Type of eyeliner | Traditional vs. Nontraditional | Traditional products were defined as any product that displayed "kajal," "kohl," or "surma" on its packaging or had been specifically described as a traditional eyeliner by its user. Nontraditional eyeliners included mascaras, eyeshadows, and all other eye cosmetics that were not labeled as kajal, kohl, or surma. |
| Type of traditional eyeliner | Kajal vs. Kohl vs. Surma | Traditional eyeliners were further evaluated by comparing the three traditional eyeliner types: "kajal," "kohl," and "surma." Labels on the eyeliner's packaging were used to determine the type of eyeliner. If no label was present on the packaging, descriptions provided by the owner were used to determine the eyeliner type. |
| Country of production | US/EU vs. Other countries vs. Afghanistan | Because the US and the EU have similarly strict lead safety guidelines for cosmetic manufacturing (see section 4.4), those made in the US and the EU (i.e., Italy, France, Germany, Spain) were compared to those made in Afghanistan and other LMICs (i.e., China, Egypt, Ethiopia, India, Morocco, Mexico, Pakistan, and Saudi Arabia). |
| Production process | Homemade vs. Manufactured | Eyeliners were considered homemade if the website they were purchased from stated explicitly that the product was homemade or if the community member who donated them stated that they made the product at home. Manufactured eyeliners were identified by examining eyeliner packaging or identifying a manufacturer on the eyeliner's website. |
| Texture | Powder vs. Cream | Texture was determined visually. Eyeliners were considered a powder if they were dry (i.e., their constituent particles separated from each other readily). Eyeliners were considered a cream if they were watery, liquid, or oily. For example, mascaras and liquid eyeliners were considered creams. |
| Labeling | "Lead-free" vs. No lead statement | Eyeliners with "lead-free" labels were evaluated to determine the veracity of the labels. |

US, United States; EU, European Union; LMICs, low- and middle-income countries.

[a] All eye cosmetics were considered eyeliners, including mascaras and eyeshadows.

If it was not possible to ascertain a characteristic, the product characteristic was denoted as "unknown."

Nilu Pedram (AHI), Aesha Mokashi (Haz Waste Program), Mohamed Ali (Haz Waste Program), and Stephen G. Whittaker (Haz Waste Program), 2023). The Afghan community provided eyeliners for analysis that were labeled as "stone surma", "oily surma", "kajal", and "kohl."

Community members revealed they often used eyeliners imported from Afghanistan or nearby countries like Pakistan. Others purchased eyeliners from local retailers.

We learned that the lack of information about the health effects and the dangers of lead in cosmetics was a prominent reason for the community's continued use of these products and resistance to change. Over 90% of the community members were unaware that traditional eyeliners contain lead. This lack of knowledge extended to the potential health risks associated with lead, with 88% of participants admitting unfamiliarity with such issues. Community members highlighted the critical need for education and awareness campaigns regarding the health risks of lead exposure. The consensus indicated a desire for more information on the health risks associated with traditional eyeliners, and they demonstrated a willingness to change purchasing behaviors if presented with safe, accessible, and culturally appropriate alternatives.

## 3.2 Eyeliner characteristics

The median lead concentration for all eyeliners in this dataset was 3.2 ppm, with a range of <0.2 to 840,000 ppm. A summary of the lead concentrations associated with product characteristics for *all eyeliners (traditional and nontraditional)* is provided in Table 3. A summary of the lead concentrations associated with product characteristics for *traditional eyeliners only* is provided in Table 4.

Traditional eyeliners contained significantly higher lead concentrations than nontraditional products ($p < 0.001$) (Table 3). The median lead concentration in traditional eyeliners was 10 ppm, compared to a median of 0.06 ppm in nontraditional products. Lead concentrations were <RL in 77% of nontraditional eyeliners compared to 20% for traditional eyeliners.

Eyeliners from Afghanistan had significantly higher lead concentrations than those made in other countries ($p = 0.016$) (Table 3). The lead concentrations in eyeliners from Afghanistan and other countries were significantly higher than those from the United States/European Union (US/EU) ($p < 0.001$, $p = 0.0011$) (Table 3). The difference in lead concentrations was reflected in the median lead concentrations for the country of production: Afghan eyeliners had a median of 29 ppm, other countries had a median of 2.8 ppm, and US/EU eyeliners had a median of 0.94 ppm (Table 3). Traditional eyeliners from Afghanistan had significantly higher lead concentrations than traditional US/EU-made eyeliners ($p = 0.019$) (Table 4).

There was no statistically significant difference in lead concentrations between manufactured eyeliners and homemade eyeliners (Table 3), nor between traditional manufactured eyeliners and traditional homemade eyeliners (Table 4). Products containing >100,000 ppm lead were associated with all production methods.

Powder eyeliners contained significantly higher lead concentrations than cream eyeliners ($p < 0.001$) (Table 3). Lead concentrations were <RL in 64% of cream eyeliners and 22% of powder eyeliners. The median lead concentration for all

Table 3. Lead concentrations in *all (traditional and nontraditional)* eyeliners by product characteristics.

| Eyeliner characteristic | n | % samples <RL[b] | Median [range] (ppm) | p-value |
|---|---|---|---|---|
| All eyeliners | 145 | 42% | 3.2 [<0.2 - 840,000] | |
| **Eyeliner type** | -- | | -- | |
| *Traditional* | 80 | 20% | 10 [<0.2 - 840,000] | Traditional > Nontraditional (<0.001) |
| *Nontraditional* | 65 | 77% | 0.06 [<0.2 - 1,800] | |
| **Country of production** | -- | | -- | Kruskal-Wallis (<0.001) |
| *US/EU* | 35 | 66% | 0.94 [<0.2 - 600,000] | Afghanistan > US/EU (<0.001) |
| *Afghanistan[a]* | 9 | 0% | 29 [1.1 - 630,000] | Afghanistan > Other (0.016) |
| *Other countries* | 60 | 37% | 2.8 [<0.2 - 840,000] | Other > US/EU (0.0011) |
| **Production process** | -- | | -- | No significant differences in lead concentrations |
| *Manufactured* | 111 | 51% | 0.67 [<0.2 - 670,000] | |
| *Homemade[a]* | 9 | 0% | 8.1 [0.9 - 720,000] | |
| **Texture** | -- | | -- | |
| *Powder* | 51 | 22% | 690 [<0.2 - 840,000] | Powder > Cream (<0.001) |
| *Cream* | 74 | 64% | 0.45 [<0.2 - 384.5] | |

RL, analytical method reporting limit; ppm, parts per million; US/EU, United States/European Union.

[a] All products in this category were traditional eyeliners.

[b] RL varied depending on the analytical method. The RL ranged from 0.05 – 4 ppm depending on the method. See S1 Appendix for more details.

**Table 4.** Lead concentrations in *traditional* eyeliners by product characteristics[a].

| Eyeliner characteristics | n | % samples <RL[b] | Median [range] (ppm) | p-value |
|---|---|---|---|---|
| All traditional eyeliners | 80 | 20% | 10 [<0.2 - 840,000] | |
| **Type of traditional eyeliner** | -- | | -- | Kruskal-Wallis (0.0021) |
| *Kohl* | 26 | 31% | 170,000 [<0.2 - 840,000] | Kohl > Kajal (0.015) |
| *Kajal* | 27 | 26% | 1.6 [<0.2 - 130,000] | Surma > Kajal (<0.001) |
| *Surma* | 20 | 5% | 1,500 [<0.2 - 670,000] | |
| **Country of production** | -- | | -- | Kruskal-Wallis (0.22) |
| *US/EU* | 10 | 20% | 1.5 [<0.2 - 600,000] | Afghanistan > US/EU (0.019) |
| *Afghanistan* | 9 | 0% | 29 [1.1 - 630,000] | |
| *Other countries* | 39 | 28% | 5.8 [<0.2 - 840,000] | |
| **Production** | -- | | -- | No significant difference in lead concentrations |
| *Manufactured* | 51 | 24% | 3.2 [<0.2 - 670,000] | |
| *Homemade* | 9 | 0% | 8.1 [0.9 - 720,000] | |
| **Texture** | -- | | -- | |
| *Powder* | 33 | 0% | 410,000 [0.5 - 840,000] | Powder > Cream (<0.001) |
| *Cream* | 40 | 38% | 1.4 [<0.2 - 384.5] | |
| **Labeling** | -- | | -- | No significant differences in lead concentrations |
| *Lead-free label* | 11 | 0% | 130,000 [0.5 - 610,000] | |
| *No lead-free label* | 31 | 0% | 8.3 [<0.2 - 720,000] | |

RL, analytical method reporting limit; ppm, parts per million; US/EU, United States/European Union.

[a]Nontraditional eyeliners were not included in this analysis.

[b]RL varied depending on the analytical method. The RL ranged from 0.05 – 4 ppm depending on the method. See S1 Appendix for more details.

powder eyeliners was 690 ppm, whereas the median for cream eyeliners was 0.45 ppm. Traditional powder eyeliners had significantly higher lead concentrations than traditional cream eyeliners (p < 0.001) (Table 4). The median lead concentration in traditional powder eyeliners was 410,000 ppm, whereas the median for traditional cream eyeliners was 1.4 ppm. No cream eyeliners contained >100,000 ppm lead.

Lead concentrations in kohl and surma were significantly higher than in kajal (p = 0.015, p < 0.001) (Table 4). The median lead concentrations in kohl and surma were 170,000 ppm and 1,500 ppm, respectively; the median concentration in kajal was 1.6 ppm. However, samples of all three types of traditional eyeliners had lead concentrations >100,000 ppm.

There was no statistically significant difference in lead concentrations between eyeliners with a lead-free label and eyeliners without a label (Table 4). Both types included eyeliners with lead concentrations ranging from below 1 ppm to >100,000 ppm. The median lead concentration in products with lead-free labels was 130,000 ppm, whereas the median concentration for products without a label was 8.3 ppm. Both types included eyeliners with lead concentrations >100,000 ppm. All eyeliners in this category were traditional eyeliners.

## 4 Discussion

### 4.1 Community engagement

Engaging the Afghan community via a CBPR approach was an effective method for gathering insights into the use of traditional eyeliners and opportunities to promote safer products. Our findings highlighted the importance of collaborative initiatives with the community to raise awareness about the potential risks of lead exposure and to encourage the adoption of safer alternatives. The use of traditional eyeliners was described as widespread and culturally significant, often associated with beliefs about its advantages for improving vision and other health benefits while also being aesthetically appealing. The overwhelming majority of responses indicated a strong preference for a safer alternative to traditional eyeliners, especially if recommended by reputable community organizations or governmental agencies that considered cultural and religious practices. Health was noted as a primary concern, and many community partners were willing to switch to a safer product that did not compromise the look and feel of traditional eyeliners. This finding aligns with previous studies demonstrating that education, accessible information, and awareness initiatives from trusted CBOs and government agencies increase the willingness to modify harmful behavior [36–38].

Many Afghan community members expressed that the primary motivator for adopting safer products and practices is the potential adverse health effects from using hazardous products on themselves and their families. It was encouraging to learn that once community members received information regarding the existence of lead in traditional eyeliners, they stated their willingness to stop using these products and advise others in their families to do the same.

### 4.2 Product terminology

A review of the terminology used to describe traditional eyeliners is presented in Table 5. The terms used to name traditional eyeliners vary based on the region and culture. "Kohl" is often used to describe traditional eyeliners in the Middle East and other majority Islamic countries [9,10,40,48] and is typically made by grinding lead-containing minerals such as galena [12,40,47,49]. "Kajal" is predominantly used to describe traditional eyeliners used in India and is often made from

**Table 5. Terminology used to describe traditional eyeliners.**

| Term[a] | Origin/ Translation | Region of use | Common ingredients | Other names |
|---|---|---|---|---|
| Surma | Urdu word for antimony [39]; Derived from the Persian word "sormeh." [40] | Afghani-stan [13], Northern India [41], Pakistan [42] | Grinding lead sulfide (PbS), also known as galena [43]; Naturally occurring ore [13]; soot from burning animal fat or almonds[b]. | Stone surma (Afghanistan) - if product is made from grinding minerals such as galena[b]; Oily surma (Afghanistan) - if product is made from soot and animal fat[b]. |
| Kajal | Arabic origin [44] | India [45], Pakistan [42] | Soot of sandalwood or other plants mixed with oil or clarified butter (ghee) [41,45,46]. | Kahal, Kanmashi, Kannu, Kappu, Kaatuka, Kanmai [15] |
| Kohl | Arabic origin [47] | Middle East [47,43,44] | Primarily galena - can be mixed with herbs (saffron, neem, fennel, etc.), gemstones, and/or marine life (coral, pearl, etc.) [47,45,46]. | Kohl stone [47] |

[a]Terms researched were based on terms our local Afghan community used for traditional eyeliners. We excluded other traditional eyeliner terms from our study (i.e., tiro, tozali, kwalli) because our local community did not use them.

[b]Information came via personal communication between Reza Pedram (AHI), Nilu Pedram (AHI), Aesha Mokashi (Haz Waste Program), Ali Mohamed (Haz Waste Program), and Stephen G. Whittaker (Haz Waste Program), 2023.

soot mixed with animal fats or vegetable oils [41,45]. The term "surma" is used in Afghanistan, North India, and Pakistan [41,42]. Some sources suggest that surma is made by grinding galena [9,13,43], while others suggest it is made from soot and animal fat [41,50]. From our community engagement, we learned that our local Afghan community uses a wide variety of traditional eyeliners irrespective of the assigned name, including "oily surma", "stone surma", "kajal", and "kohl".

It is noteworthy that many original eyeliner recipes called for grinding antimony/stibnite ($Sb_2S_3$) to a fine powder. However, the rarity of this element has prompted many producers to use galena, which is less expensive and easier to obtain [12,14].

## 4.3 Product composition

Our finding of very high lead concentrations in some traditional eyeliners is consistent with previous studies (Table 6). McMichael and Stoff (2018) analyzed ten samples of Afghan-produced traditional eyeliner and determined that all samples contained lead, with concentrations ranging from 400 to 830,000 ppm [13]. Studies conducted in the Middle East found lead in traditional eyeliners ranging from below the limit of detection to 973,000 ppm [6–9,11]. Researchers in Europe found similar results, with some traditional eyeliners containing very low lead levels and others containing thousands of ppm [10,12]. New York City's Department of Health and Mental Hygiene has tested various cosmetics since 2011, including traditional eyeliners from in-home investigations and store surveys, and found multiple products containing >100,000 ppm [4]. Despite the US Food and Drug Administration's (FDA) Import Alert, we found that traditional eyeliners containing >100,000 ppm lead continue to be imported to the US (examples are provided in Fig 1).

Several researchers have also studied lead in nontraditional eye cosmetics, including eyeliners, eyeshadows, and mascaras (Table 7). While lead concentrations in nontraditional eye cosmetics were typically much lower than those in traditional products, some still contained hundreds to thousands of ppm lead.

## 4.4 Factors associated with high lead concentrations

Our analysis of eyeliner characteristics associated with high concentrations revealed no statistically significant differences between homemade vs. manufactured products and those labeled lead-free vs. no labeling. However, the following factors were associated with higher concentrations: traditional products (compared to nontraditional), the terms "kohl" or "surma" (compared

**Table 6. Lead concentrations in traditional eyeliners measured by different studies.**

| Study | Country | Cosmetic term used | Lead range (ppm) |
|---|---|---|---|
| Current study | US | Kajal, Kohl, Surma | <0.2 - 840,000 |
| Hore & Sedlar 2024 [51] | US | Kohl, Kajal, Surma | <0.33 - 980,000 |
| Lead content of consumer products tested in King County, WA (2023) [2] | US | Kohl, Kajal, Surma | 390,000 - 840,000 |
| Navarro-Tapia et al. 2021 [12] | Spain | Kohl-based eye cosmetics | 1.73 - 410,806 |
| Rasheed et al. 2021 [11] | Iraq | Kohl | 0.727 - 1,491 |
| Buksh et al. 2020 [9] | Pakistan | Kohl | <2.32 - 199.9 |
| Filella et al. 2020 [10] | Switzerland | Kohl | <20 - 467,632 |
| McMichael & Stoff 2018 [13] | Afghanistan | Surma | 400 - 830,000 |
| Daar et al. 2017 [8] | Jordan | Kohl | 30,000 - 570,000 |
| Gouitaa et al. 2016 [7] | Morocco | Kohl | 1 - 973,800 |
| Hardy et al. 2002 [6] | United Arab Emirates | Kohl | <50,000 - 970,000 |

ppm, parts per million; WA, Washington State; US, United States.

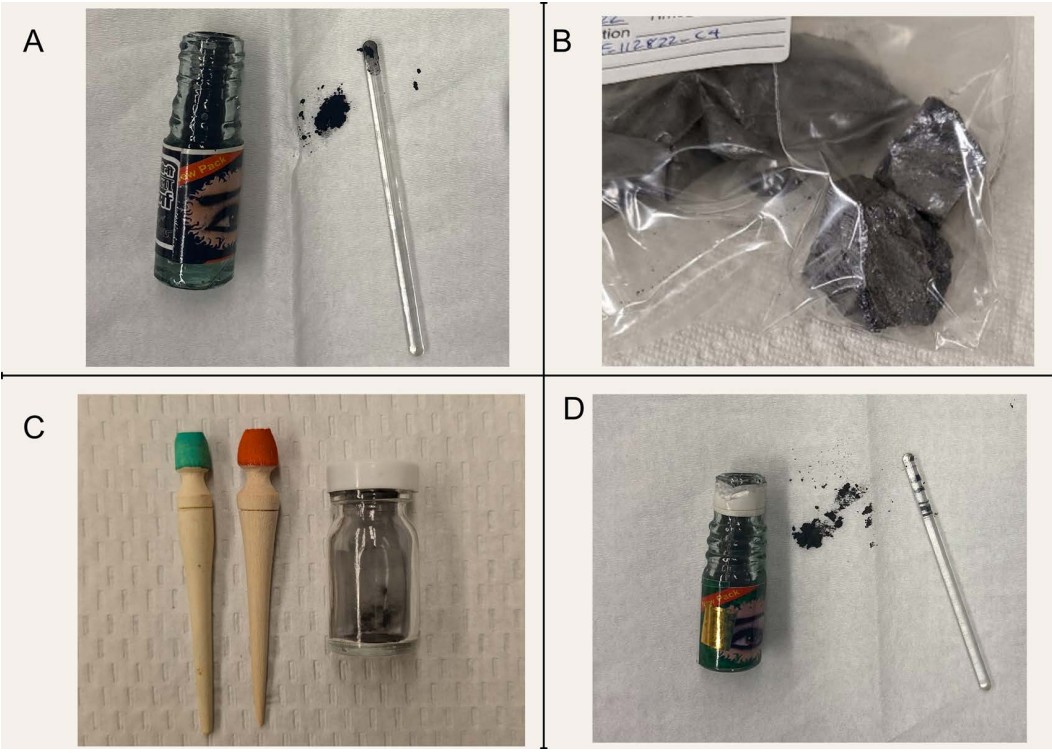

**Fig 1. Eyeliners available for purchase within the US with >100,000 ppm lead.** These eyeliners were purchased online and shipped to the US. They were then sent for laboratory analysis, and we were provided the following lead concentrations for each product: A) Surma: 140,000 – 270,000 ppm B) Galena: 393,000 ppm C) Kohl: 720,000 ppm D) Surma: 130,000 ppm. All photos taken by the authors.

**Table 7. Lead concentrations in nontraditional eyeliners measured by different studies.**

| Study | Country | Cosmetic type | Lead range (ppm) |
|---|---|---|---|
| Current study | US | Eyeliners, Eyeshadows, Mascaras | <0.2 - 1,800 |
| Metal Content of Consumer Products Tested by the NYC Health Department (2023) [4] | US | Eyeliners, Eyeshadows | <0.36 - 930,000 |
| Lead content of consumer products tested in King County, WA (2023) [2] | US | Eyeliners, Eyeshadows | 15 - 100 |
| Pawlaczyk et al. 2021 [52] | Poland | Eyeshadows | 0.0957 - 15.95 |
| Lim et al. 2018 [53] | South Korea | Eyeliners | <0.02 - 9.71 |
| Omenka et al. 2016 [54] | Nigeria | Eyeliners | 3.71 - 27.5 |
| Volpe et al. 2012 [55] | Italy | Eyeshadows | Chinese brands: 9.53 - 65.6<br>US brands: 0.25 - 2.75<br>Italian brands: 0.57 - 7.64 |
| Charter et al. 2011 [56] | Canada | Eyeliners, Eyeshadows | <0.04 - 110 |
| Omolaoye et al. 2010 [57] | Nigeria | Eyeshadows | <0.5 - 46.67 |
| Al-Saleh et al. 2009 [5] | Saudi Arabia | Eyeshadows | 0.42 - 58.72 |
| Sainio et al. 2000 [58] | Finland | Eyeshadows | <0.5 - 16.8 |

ppm, parts per million; NYC, New York City; WA, Washington State; US, United States.

to "kajal"), manufactured in Afghanistan and other countries (compared to the US and EU), and powdered products (compared to creams). Hardy et al. (2002) and Filella et al. (2020) also noted that higher lead concentrations were typically associated with powdered products [6,10]. These investigators found that traditional powdered eyeliners were often composed of a lead sulfide compound (likely galena). However, creams or liquid traditional eyeliners were typically composed of carbon compounds, silicone compounds, and waxes. Hore et al. (2024) also observed that kohl and surma contained higher lead concentrations than kajal, further indicating that surma and kohl are galena-based, whereas kajal is comprised primarily of soot [51]. Despite having lower lead concentrations, soot-based eyeliners have been found to contain polycyclic aromatic hydrocarbons (PAHs), a carcinogenic class of compounds formed from the incomplete combustion of organic matter [59,60].

While the US and EU have enforced strong cosmetic safety guidelines on lead, other regions, particularly low- and middle-income countries (LMICs), have less strict guidelines (or no guidelines) and may have less oversight in eyeliner production, resulting in eyeliners with higher lead concentrations (Table 8).

## 4.5 Strengths and limitations of the study

A major strength of this study was our engagement with the local Afghan community, who provided considerable insights into their use of traditional and nontraditional eyeliners. Our CBPR approach facilitated the gathering of crucial information concerning terminology and the religious and cultural basis for the use of these products.

**Table 8. Safety guidelines for lead concentration in eyeliners in different countries.**

| Country/ Region | Lead limit (ppm)[a] | Specific to eyeliners? | Safety guideline | Comments |
|---|---|---|---|---|
| Federal (US) | 10 | No. The safety guideline refers to lip products and any externally applied cosmetics. | Lead in Cosmetic Lip Products and Externally Applied Cosmetics: Recommended Maximum Level | 10 ppm is the recommended maximum level of lead in cosmetic products by the US FDA (as of June 2023) [61]. FDA Import Alert 53–06: The FDA issued an Import Alert on "Kohl, Kajal, Al-Kahal, Surma, Tiro, Tozali, or Kwalli" (as of April 2023) [62]. |
| Washington state (US) | 1 | No. The bill refers to any cosmetics as defined in RCW 69.04.011 [63]. | Toxic Free Cosmetics Act | This bill restricts lead or any lead compounds intentionally added or present at 1 ppm or higher in any cosmetic (goes into effect in 2025) [64]. |
| EU | 0 | No. The safety guideline refers to all cosmetics. | (EC) No 1223/2009 (Article 14, Annex 2) | "Non-intended presence of a small quantity of a prohibited substance (such as lead and its compounds) is permitted provided that it follows certain requirements" (as of August 2023) [65]. |
| Germany | 5 | Yes. The safety guideline refers to kajal, eye shadow, eyeliner, makeup powder, rouge, and theater, fan, or carnival makeup. | Technically Avoidable Heavy Metal Contents in Cosmetic Products | Heavy metal content in traditional eyeliners exceeding 5 ppm is considered technically avoidable (as of August 2023) [66]. |
| Canada | 10 | No. The safety guideline refers to any cosmetic. | Guidance on Heavy Metal Impurities: Section 4 | Lead content above 10 ppm is considered "technically avoidable" (as of August 2023) [67]. |
| India[a] | 20 | No. The safety guideline applies to any ingredient meant to add color to a cosmetic. | Rule 134 of the Drugs and Cosmetics Act and Rules | The rule states that "the permitted synthetic organic colors and natural organic colors used in the cosmetic shall not contain more than 20 ppm lead" (as of August 2023) [68]. |
| Saudi Arabia[a] | 10 | No. The safety guideline refers to all cosmetics and personal care products. | NO. SFDA.CO/GSO 1943:2021 | "Non-intended and technically unavoidable levels of lead are permitted in cosmetics and personal care items if they do not exceed 10 ppm" (as of April 2024) [69]. |
| Pakistan[a] | No safety guidelines | -- | -- | -- |
| Afghanistan | No safety guidelines | -- | -- | -- |

ppm, parts per million; US, United States; FDA, Food and Drug Administration; EU, European Union.

[a]India, Saudi Arabia, and Pakistan were included because we found that the local Afghan community uses traditional eyeliners made in these countries.

We also learned about the preferred safer alternatives already used by local Afghans, which allowed us to promote safer products to others in the community. In addition, acquiring traditional and nontraditional eyeliners from multiple sources provided a representative sample of products used by our local communities.

Limitations include our inability to determine the relative contribution of eyeliners to Afghan children's overall lead exposure. As stated in our previous paper that focused on lead in cookware [70], we recognize that these children may have suffered high lead exposures in Afghanistan. In addition, the in-home investigations identified other lead-containing items, including aluminum cookware, glazed dishes, silverware, spices, and cosmetic jewelry (Personal communication between Sharon G. Cohen (Haz Waste Program) and Stephen G. Whittaker (Haz Waste Program), 2021). Our ability to determine the effectiveness of removing sources of lead exposure is also compromised by the lack of follow-up BLL testing by some healthcare providers (Personal communication between Sharon G. Cohen (Haz Waste Program) and Stephen G. Whittaker (Haz Waste Program), 2021).

Another limitation concerns the chemical analysis of eyeliner products. It was often necessary to collect at least 1 g of sample to achieve RL below 1 ppm because of matrix interference. However, collecting 1 g was frequently not possible when community members provided limited quantities of products. In addition, because many samples were analyzed before we recognized the limitations in analytical methods, the method chosen was not consistent and not always optimal. For example, when using ICP-MS and GFAA, the typical RL was 0.05 ppm and 0.2 ppm, respectively for 1 g of sample, whereas ICP-OES had a higher typical RL of 4 ppm. This resulted in considerable differences in the number of samples with data <RL. No samples analyzed by ICP-MS were reported to be < RL, whereas 12% of samples analyzed by GFAA were <RL, and 78% of samples analyzed by ICP-OES were <RL. Consequently, we were compelled to employ imputation methods, and we took steps to present data in a comparable manner. However, a positive outcome of navigating the challenges of multiple analytical methods was that it allowed PHSKC to develop a standardized analytical protocol for consumer products using one consistent analytical method.

## 5 Conclusions and recommendations

To our knowledge, this is the first study to combine a CBPR strategy with comprehensive chemical analysis to describe the use of and hazards associated with traditional eyeliners in the US. We learned that deeply held cultural beliefs frequently underpin the use of eyeliner. Nonetheless, with the insights provided by our local Afghan community, we identified safer cosmetics that are readily available in the US and acceptable to many in this community. With this information, we will be able to facilitate community-led interventions to promote safer cosmetics in our local communities.

Our study demonstrated that newcomers from Afghanistan are not the only communities at risk. These eyeliners are manufactured and distributed worldwide, where they are used by multiple communities for religious and cultural purposes. This is a particular concern for children, where recent data suggest that one in three suffers from lead poisoning worldwide, with a disproportionate burden on children in LMICs [71–73].

A review of current safety guidelines limiting the amount of lead in cosmetics revealed that many countries have promulgated strict limits. However, this study, in addition to previous investigations, demonstrates the ready availability of lead-containing eyeliners available for purchase in the US and elsewhere [2,4].

We also learned that the terms "surma," "kohl," and "kajal" are used interchangeably in the scientific literature and between cultures and regions [9,48,74]. Therefore, although some products may be made from soot, it is not possible to distinguish these products from those made from ground galena based on their assigned name. Further, the designation of an eyeliner as "nontraditional" or "lead-free" does not guarantee that the product is lead-free.

### 5.1 Recommendations

This study's findings highlight the need to prevent the use of lead-containing minerals to manufacture eyeliners and other cosmetics. Although safety guidelines limiting the lead content in eyeliners exist in many countries, these guidelines

are poorly enforced. This is especially true in the US, where the FDA enforces lead safety guidelines and has issued an Import Alert for many traditional products. However, given that we readily purchased lead-containing eyeliners in King County, WA, the FDA should expand its enforcement actions to further curb the importation of these products and prevent their sale by brick-and-mortar stores and online retailers.

Special attention should be paid to resettled children, who are especially prone to adverse health effects due to their forced displacement, experiences with conflict, and statelessness [75]. Several studies have reported that refugee children have a higher prevalence of cases that exceed the CDC's BLRV compared to the general population [76]. Children from LMICs with lead poisoning exhibit a greater loss of intelligence quotient and greater incidence of mild intellectual disabilities compared to children from higher-income countries [77,78].

It is noteworthy that we became aware of lead exposure among Afghan children because the CDC recommends a medical screening for refugees as they enter the US. However, for other children in WA, only 4.4% of those 72 months of age and younger had BLL tests in 2016, compared to the corresponding national testing rate of 17.0% [79]. This relatively low testing rate is a significant impediment to lead poisoning prevention interventions in WA. Consequently, BLL testing should be conducted on all Medicaid-eligible children at 12 months of age and then again at 24 months of age. It is also vital that all children previously determined to have BLLs at or above the CDC's BLRV undergo follow-up testing by their healthcare providers [79].

Efforts should also be devoted to engaging with cultural and religious leaders to further understand whether the use of lead-containing eyeliners is required for compliance with traditional practices. It is vital to understand whether the use of lead-containing minerals is an absolute requirement or whether a safer alternative yielding the same appearance would be acceptable. These leaders could then be engaged in a community-driven campaign to promote safer eyeliners.

As recommended by Alex-Oni et al. (2023), we need a national repository of consumer product lead surveillance data [80]. Such a national database would enhance our ability to identify populations at risk and formulate effective interventions. To our knowledge, publicly accessible databases describing lead in consumer products in the US are only available from New York City and King County, WA [2,4]. Federal resources should be devoted to funding data gathering in other jurisdictions and the information consolidated into a single data repository.

Other recommendations reflect those we have made previously when considering the ready availability of lead-containing cookware in the US [70,81]. These include: providing information about lead in products to newcomer communities in a culturally appropriate manner; developing community-informed intervention strategies that involve substituting lead-containing products with safer alternatives; sharing information with all US residents about the presence of lead in products; educating healthcare providers about the risks to children associated with BLLs at or near the BLRV; and providing sufficient resources to local health jurisdictions to effectively manage blood lead surveillance data and conduct interventions with at-risk children.

## Supporting information

**S1 Appendix. Provides detailed information on sample acquisition and laboratory analysis methods.**
(DOCX)

**S2 Appendix. Provides regression analysis results and reporting parameters.**
(DOCX)

**S3 Appendix. Provides the data used to conduct analysis on the eyeliner characteristics (Type, Country of production, Texture, Production process, Labels).**
(CSV)

## Acknowledgments

We gratefully acknowledge the contributions of the Afghan community in King County, Washington (WA), to this work. Nilu Pedram and Reza Pedram (Afghan Health Initiative) provided valuable insights into Afghan culture, and Nilu Pedram conducted in-home surveys and product collection. We are also thankful to Public Health - Seattle & King County for sponsoring this project, and specifically to the Residential Services Program within the Hazardous Waste Management Program in King County, WA, which conducted the in-home investigations. The Residential Services Program and the Environmental Health Community Toxics Section organized and staffed the community product lead testing events. We would also like to thank all the community-based organizations that partnered with us for the community product lead testing events. The University of Washington's Environmental Health Laboratory and Trace Organics Analytical Center (Seattle, WA) (UW EHL TOAC) and NVL Laboratories, Inc. (Seattle, WA) provided technical expertise throughout this project. Shar Samy (UW EHL TOAC), Nick Ly (NVL), and Shalini Patel (NVL) provided insights into eyeliner product testing. Cynthia Wu (University of Washington) provided advice on statistical analysis.

## Author contributions

**Conceptualization:** Aesha Mokashi, Katie M. Fellows, Roger J. Chin, Stephen G. Whittaker, Diana M. Ceballos.

**Formal analysis:** Aesha Mokashi, Roger J. Chin.

**Investigation:** Aesha Mokashi, Katie M. Fellows, Roger J. Chin.

**Methodology:** Katie M. Fellows, Roger J. Chin, Stephen G. Whittaker, Diana M. Ceballos.

**Writing – original draft:** Aesha Mokashi, Roger J. Chin, Stephen G. Whittaker.

**Writing – review & editing:** Aesha Mokashi, Katie M. Fellows, Roger J. Chin, Stephen G. Whittaker, Christopher D. Simpson, Diana M. Ceballos.

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
