## [Decision Letter · Decision Letter 0]

PGPH-D-25-00103

Lead in traditional eyeliners: An investigation into use and sources of exposure in King County, Washington

Dear Dr. Ceballos,

Thank you for submitting your manuscript to PLOS Global Public Health. After careful consideration, we feel that it has merit but does not fully meet PLOS Global Public Health’s publication criteria as it currently stands. Therefore, we invite you to submit a revised version of the manuscript that addresses the points raised during the review process.

We look forward to receiving your revised manuscript.

Kind regards,

Raquel Muñiz-Salazar, Ph.D.

Academic Editor

Journal Requirements:

2. We do not publish any copyright or trademark symbols that usually accompany proprietary names, eg (R), (C), or TM (e.g. next to drug or reagent names). Please remove all instances of trademark/copyright symbols throughout the text, including ™ on page 11.

3. Figure 1: Please confirm (a) that you are the photographer; or (b) provide written permission from the photographer to publish the photo(s) under our CC-BY 4.0 license.

4. Figure 1  contains branding/a logo. We are not permitted to publish this under our CC-BY 4.0 license, even with permission. We ask that you please remove or replace it.

Additional Editor Comments (if provided):

This study highlights a significant public health issue by identifying hazardous lead levels in traditional eyeliners used by refugee and immigrant communities. The research underscores the need for stronger regulations and community education to reduce exposure and protect vulnerable populations, thereby helping to decrease health disparities.

The reviewers have provided minor suggestions for improvement.

Reviewers' comments:

Reviewer's Responses to Questions

**Comments to the Author**

1. Does this manuscript meet PLOS Global Public Health’s publication criteria ? Is the manuscript technically sound, and do the data support the conclusions? The manuscript must describe methodologically and ethically rigorous research with conclusions that are appropriately drawn based on the data presented.

Reviewer #1: Yes

Reviewer #2: Yes

2. Has the statistical analysis been performed appropriately and rigorously?

Reviewer #1: Yes

Reviewer #2: Yes

3. Have the authors made all data underlying the findings in their manuscript fully available (please refer to the Data Availability Statement at the start of the manuscript PDF file)?

Reviewer #1: Yes

Reviewer #2: Yes

4. Is the manuscript presented in an intelligible fashion and written in standard English?

Reviewer #1: Yes

Reviewer #2: Yes

5. Review Comments to the Author

Reviewer #1: I have thoroughly reviewed your manuscript and found it to be a well-conducted and valuable study. Below, you will find my detailed evaluation and comments.

Technical Soundness and Data Support for Conclusions:

The manuscript is technically sound with a well-designed methodology that combines quantitative and qualitative approaches. The research employs:

Community-based participatory research (CBPR) with the Afghan community through surveys and community forums

Chemical analysis of 145 eyeliner samples using established analytical methods (ICP-MS, ICP-OES, GFAA)

Statistical analysis to identify factors associated with lead concentrations

Review of relevant regulations and guidelines across multiple countries

The conclusions are appropriately drawn from the data presented. For example, the finding that traditional eyeliners contained significantly higher median lead concentrations (10 ppm) than nontraditional eyeliners (0.06 ppm) is directly supported by the chemical analysis results. Similarly, the conclusion about community members' lack of awareness about lead risks is backed by survey data showing over 90% were unaware of lead content in traditional eyeliners.

Statistical Analysis:

The statistical analysis is rigorous and appropriate for the data type and research questions. Specifically:

Non-parametric tests (Wilcoxon Rank-Sum and Kruskal-Wallis) were correctly chosen given the non-normal distribution of the data

Proper handling of censored data (values below reporting limits) using established methods through the NADA2 and EnvStats packages

Adequate sample sizes were ensured (minimum of 6 samples per variable) for statistical power

Clear reporting of statistical significance (p-values) and effect sizes

Appropriate bivariate and multivariate analyses to examine relationships between variables

Data Availability:

The authors have made their data fully available through multiple channels:

All product testing data is publicly accessible through the "Lead Content of Consumer Products tested in King County, Washington" open data platform

Detailed supplementary materials are provided:

S1 Appendix: Methods details

S2: Statistical analysis parameters and results

S3: Complete eyeliner characteristics dataset

Raw data points behind summary statistics are available

The Data Availability Statement clearly outlines how to access all data

Presentation and Language:

The manuscript is exceptionally well-written and structured. It features:

Clear and logical organization with appropriate sections

Professional academic English throughout

Well-designed tables and figures that effectively communicate key findings

Precise technical language while remaining accessible

Proper citation and attribution of sources

Clear explanations of technical terms and cultural concepts

My additional comments on the manuscript are as below:

Strengths:

Strong ethical considerations with proper IRB documentation

Excellent integration of community engagement with scientific analysis

Comprehensive literature review and regulatory analysis

Clear public health implications and actionable recommendations

The level of detail in the supplementary statistical information enhances the reproducibility of the study and demonstrates thorough statistical methodology, which is particularly important for public health research of this nature.

Minor suggestions for improvement:

If only other reviewers requested too, some method details in the supplementary materials could be moved to the main text.

Although it is not final form, in the reviewer copy of the manuscript, the figure 1 is in low resolution. Figure 1 could benefit from higher resolution images.

Reviewer #2: The manuscript is well-written, interesting to read, used rigorous statistical methods for data analysis etc. While the presence of lead in cosmetic products, especially eyelines has been discussed in the literature for some time, the comprehensive approach, using both qualitative and quantitate methods makes this research a valuable contribution to the body of research.

Some minor comments:

Line 36 - "aged" instead of "ages"

Line 174 - "eyeliners that were suspected to contain lead" - please provide details on why they were suspected, on what basis?

Section 2.5 Sample analysis - The authors have used 3 different types of chemical analysis of the samples - ICP MS, ICP OES and GF AAS. What was the rationale for using 3 different types? Especially that the methods have various RLs and it ended up being a limitation for the research.

Line 214 - here and in several other places the authors speak about the RLs that the lead concentrations in samples were checked against. They provide different RLs in Table 8 and report the percentage of samples <rl 3="" 4.="" and="" however="" in="" tables="">

Thank you.</rl>

6. PLOS authors have the option to publish the peer review history of their article (what does this mean? ). If published, this will include your full peer review and any attached files.

**Do you want your identity to be public for this peer review?** For information about this choice, including consent withdrawal, please see our Privacy Policy .

Reviewer #1: **Yes: ** ONUR KENAN ULUTAŞ

Assoc. Prof.

B.Pharm; MSc., PhD in Toxicology; AAS- Design&Coding

Reviewer #2: No

---

## [Decision Letter · Decision Letter 1]

Lead in traditional eyeliners: An investigation into use and sources of exposure in King County, Washington

PGPH-D-25-00103R1

Dear Dr Ceballos,

We are pleased to inform you that your manuscript 'Lead in traditional eyeliners: An investigation into use and sources of exposure in King County, Washington' has been provisionally accepted for publication in PLOS Global Public Health.

Best regards,

Raquel Muñiz-Salazar, Ph.D.

Academic Editor

Following a thorough review of the revised manuscript, we confirm that all reviewers' comments have been satisfactorily addressed.

We are pleased to inform you that the final decision is to accept your manuscript for publication.

Reviewer Comments (if any, and for reference):

Reviewer's Responses to Questions

**Comments to the Author**

1. If the authors have adequately addressed your comments raised in a previous round of review and you feel that this manuscript is now acceptable for publication, you may indicate that here to bypass the “Comments to the Author” section, enter your conflict of interest statement in the “Confidential to Editor” section, and submit your "Accept" recommendation.

Reviewer #2: All comments have been addressed

2. Does this manuscript meet PLOS Global Public Health’s publication criteria ? Is the manuscript technically sound, and do the data support the conclusions? The manuscript must describe methodologically and ethically rigorous research with conclusions that are appropriately drawn based on the data presented.

Reviewer #2: Yes

3. Has the statistical analysis been performed appropriately and rigorously?

Reviewer #2: Yes

4. Have the authors made all data underlying the findings in their manuscript fully available (please refer to the Data Availability Statement at the start of the manuscript PDF file)?

Reviewer #2: Yes

5. Is the manuscript presented in an intelligible fashion and written in standard English?

Reviewer #2: Yes

6. Review Comments to the Author

Reviewer #2: (No Response)

7. PLOS authors have the option to publish the peer review history of their article (what does this mean? ). If published, this will include your full peer review and any attached files.

**Do you want your identity to be public for this peer review?** For information about this choice, including consent withdrawal, please see our Privacy Policy .

Reviewer #2: No
